# Investigating the ROS Formation and Particle Behavior of Food-Grade Titanium Dioxide (E171) in the TIM-1 Dynamic Gastrointestinal Digestion Model

**DOI:** 10.3390/nano15010008

**Published:** 2024-12-25

**Authors:** Nicolaj S. Bischoff, Anna K. Undas, Greet van Bemmel, Jacco J. Briedé, Simone G. van Breda, Jessica Verhoeven, Sanne Verbruggen, Koen Venema, Dick T. H. M. Sijm, Theo M. de Kok

**Affiliations:** 1Department of Translational Genomics, GROW Research Institute for Oncology and Reproduction, Maastricht University Medical Centre, 6200 MD Maastricht, The Netherlands; j.briede@maastrichtuniversity.nl (J.J.B.); s.vanbreda@maastrichtuniversity.nl (S.G.v.B.); t.dekok@maastrichtuniversity.nl (T.M.d.K.); 2Wageningen Food Safety Research (WFSR), Wageningen University & Research, 6708 WB Wageningen, The Netherlands; anna.undas@wur.nl (A.K.U.); greet.vanbemmel@wur.nl (G.v.B.); 3Centre for Healthy Eating & Food Innovation, Maastricht University—Campus Venlo, Villafloraweg 1, 5928 SZ Venlo, The Netherlandss.verbruggen@maastrichtuniversity.nl (S.V.); koen.venema@wur.nl (K.V.); 4Department of Pharmacology and Toxicology, Maastricht University, 6229 ER Maastricht, The Netherlands; dick.sijm@maastrichtuniversity.nl; 5Office of Risk Assessment and Research, Netherlands Food and Consumer Product Safety Authority, P.O. Box 43006, 3540 AA Utrecht, The Netherlands

**Keywords:** E171, titanium dioxide, human gastrointestinal digestion, agglomeration, ROS, physicochemical characterization

## Abstract

Food-grade titanium dioxide (E171) is widely used in food, feed, and pharmaceuticals for its opacifying and coloring properties. This study investigates the formation of reactive oxygen species (ROS) and the aggregation behavior of E171 using the TNO Gastrointestinal (GI) model, which simulates the stomach and small intestine. E171 was characterized using multiple techniques, including electron spin resonance spectroscopy, single-particle inductively coupled plasma–mass spectrometry, transmission electron microscopy, and dynamic light scattering. In an aqueous dispersion (E171-aq), E171 displayed a median particle size of 79 nm, with 73–75% of particles in the nano-size range (<100 nm), and significantly increased ROS production at concentrations of 0.22 and 20 mg/mL. In contrast, when E171 was mixed with yogurt (E171-yog), the particle size increased to 330 nm, with only 20% of nanoparticles, and ROS production was inhibited entirely. After GI digestion, the size of dE171-aq increased to 330 nm, while dE171-yog decreased to 290 nm, with both conditions showing a strongly reduced nanoparticle fraction. ROS formation was inhibited post-digestion in this cell-free environment, likely due to increased particle aggregation and protein corona formation. These findings highlight the innate potential of E171 to induce ROS and the need to consider GI digestion and food matrices in the hazard identification/characterization and risk assessment of E171.

## 1. Introduction

Titanium dioxide (TiO_2_) is a white food coloring and opacifying agent known as E171 in the European Union and Asia and INS171 in North America. E171 is widely used in various products, including candies, confectionaries, sauces, and pharmaceuticals [1,2,3]. While E171 has been historically deemed safe for oral intake, it has come under recent scrutiny due to emerging evidence of its potential genotoxic effects, alterations in the gut microbiome, and pro-inflammatory properties [1,4]. This shift in perception of its safety has led the European Food Safety Authority (EFSA) to re-evaluate E171 and declare it unsafe as a food additive [3]. Subsequently, this resulted in the ban of E171 as a food additive within the European Union as of August 2022 and its removal from Annexes II and III of Regulation (EC) No 1333/2008 (Commission Regulation 2022/63) [5]. However, other countries such as the USA, the UK, or China still allow it at dose levels that are currently considered safe.

Initially, concerns have been raised by studies indicating E171’s potential role in colorectal cancer (CRC). Research utilizing rodent models, particularly those exposed to azoxymethane (AOM) and dextran sodium sulfate (DSS) to induce colitis-associated cancer, demonstrated an upregulation of tumor progression markers such as COX2, Ki67, and beta-catenin. These studies also observed dysplastic changes in the colonic epithelium, including goblet cell reduction [6,7]. Additionally, E171 has been linked to the development of aberrant crypt foci, preneoplastic lesions in the colon, and immune system alterations, evidenced by shifts in Th1/Th17 balance [6]. Toxicogenomic studies further corroborate these findings, showing transcriptional changes in colonic tissues in mice and human colon carcinoma cell lines indicative of early markers of CRC development [8,9,10,11]. Furthermore, recent studies from Putra et al. (2024) in human feces raised concerns due to increased markers for gut permeability and neutrophil inflammation in young healthy adults [12].

Despite EFSA’s recommendation for E171’s withdrawal, contrasting viewpoints exist. For instance, regulatory bodies like Health Canada and the Food and Drug Administration (FDA) in the United States of America (USA) maintain their position that E171 is safe. The FDA and Health Canada argue that its interaction with food and the digestive tract may mitigate its adverse effects, as demonstrated in in vitro and animal models, through increased particle aggregation within the food matrix and along the gastrointestinal (GI) route [13,14,15].

The interaction between the food matrix, GI digestion, and particle behavior, particularly for E171, remains underexplored. Recent studies have demonstrated significant physicochemical changes in E171 during simulated GI digestion, such as alterations in aggregation, agglomeration, and protein corona formation [16,17,18]. These changes have implications for particle behavior and reactivity, as demonstrated by Ersöz et al. (2022), who observed increased particle aggregation and reduced cell viability but decreased intracellular reactive oxygen species (ROS) production when TiO_2_ nanoparticles were digested in milk-based matrices [19]. Other studies showed a decrease in cell viability and increases in ROS formation and genotoxicity following exposure to physiologically relevant doses of various digested TiO_2_ nanomaterials in intestinal cell lines [20,21]. While these studies identified significant alterations to the size and aggregation status of TiO_2_ nanoparticles, the actual effect of GI digestion on aggregation might be overestimated, as shown by Ferraris et al. (2023) [16]. This study demonstrated that dynamic light scattering (DLS) measurements were unsuitable for adequately assessing the aggregate size and overestimated the aggregation state following in vitro digestion. Ferraris et al. (2023) proposed spICP-MS as a more accurate and reliable method to determine particle size and state of aggregation. This approach is further supported by the emergence of validated protocols for spICP-MS evaluation specifically designed for E171 from the Joint Research Centre (JCR) [22]. Nevertheless, a comprehensive understanding of the effects of prolonged and dynamic GI digestion on E171, particularly on its ROS-generating capabilities, is still lacking.

Our research aimed to investigate these shortcomings by deploying the TNO GI Model (TIM), developed by the Netherlands Organization for Applied Scientific Research (TNO) [23]. We used the validated TIM-1 system to replicate the conditions of the upper GI tract, including the stomach and small intestine, over six hours. Our assessment focused on how GI digestion influences the characteristics of E171 particles and their ROS-generating capabilities. This study furthermore aims to bridge the knowledge gap between in vitro and human digestion since we are simultaneously conducting a human dietary intervention cross-over study, which uses the same yogurt food matrix to administer E171 to a group of healthy volunteers.

## 2. Materials and Methods

### 2.1. Determination of a Physiologically Relevant E171 Dose

We used the 2021 exposure assessment of E171, published by EFSA, to determine a physiologically relevant dose for adults (18–64 years). In the non-brand-loyal scenario of the refined exposure assessment scenario by EFSA for adults, a mean exposure of 0.3–3.8 mg/kg_bw_/day was estimated. Therefore, we choose an average daily exposure of 2.0 mg/kg_bw_/day. According to EFSA, a realistic body weight for adults is 70 kg [24]. Hence, our estimated physiologically relevant dose of E171 was determined as 140 mg/day. European countries vary in their daily dairy consumption recommendations, ranging between 153 and 481 g of dairy products daily [25]. We chose a daily consumption of 210 g of yogurt, resulting in a final concentration of 0.66 mg of E171 per 1 g of yogurt. These consequent choices were aligned with a human dietary intervention study at Maastricht University to compare this in vitro study and the human situation.

### 2.2. E171-aq and E171-yog Sample Preparation

The Sensient Technologies Company in Mexico kindly donated E171. We assessed E171 under two conditions: a fasted state, where E171 was solely dispersed in MilliQ water, and a fed state, where we mixed E171 with yogurt. E171 stock dispersion was prepared at 20 mg/mL in MilliQ water and sonicated for 16 min at 37 kHz in a water bath before each experiment. This dispersion represented the fasted state (E171-aq).

We obtained biological yogurt from the local T’Bakhuis (Ulvend, Belgium) dairy producer and sourced it fresh before each experiment. The yogurt was tested for TiO_2_ background levels and showed no contamination with TiO_2_. E171–yogurt (E171-yog) was prepared by precooling a Springlane Erika ice cream machine (Springlane GmbH, Düsseldorf, Germany) for 10 min until it reached −37 °C. We added 7 mL of the previously dispersed E171-aq (20 mg/mL) to 210 g of yogurt, resulting in 140 mg of E171 per portion of yogurt and a final concentration of 0.66 mg E171/g yogurt. The E171–yogurt mixture was stirred in the precooled mixer for 10 min before being portioned into small plastic containers (Paardekooper, Aalsmeer, The Netherlands) and stored at 4 °C until experiments or at −20 °C until analysis.

### 2.3. Physicochemical Characterization of E171-aq and E171-yog

E171 particle size and particle size distributions were determined via spICP-MS, Nexion 2000 (Perkin Elmer, Norwalk, CT, USA), and transmission electron microscopy (TEM, JEM-1400Plus, JEOL, Tokyo, Japan). We additionally analyzed the state of the aggregation’s hydrodynamic diameter (HD) and the surface charge (zeta-potential) using a Malvern NanoZS (Malvern Instruments, Malvern, UK).

Samples containing pristine E171 were characterized following dispersion according to the NanoGenotox Dispersion protocol, with minor adaptations. In short, samples were pre-wetted with 70% ethanol, dispersed in 0.05% bovine serum albumin (BSA) water, and sonicated at 37 kHz for 16 min [26]. The E171 dispersion was diluted to obtain suitable concentrations for the spICP-MS, according to the single-particle rule and as described before [22]. Instrument details and measurement setup are described in Appendix A. TEM grids were also prepared according to Verleysen et al. (2020), describing a standardized and validated method for the physicochemical characterization of pristine E171 food additives on AGS162-4 Formvar carbon-coated 400-mesh copper grids (Agar Scientific, Stansted, Essex, UK) for electron microscopy (EM, Agar Scientific, Essex, UK). Fifty images were taken, and particle size and particle size distributions were determined using the ImageJ plug-in ParticleSizer [22,27]. DLS measurements for HD and the zeta potential were conducted at a concentration of 50 µg/mL, with the temperature set to 25 °C, equilibration time of 0 s, viscosity of 0.8872 cP, and a refraction index of 1.330. E171-yog was analyzed using the same experimental conditions.

### 2.4. Measurement of ROS Formation via Electron Spin Resonance (ESR) Spectroscopy of E171-aq and E171-yog

ESR is a method that can directly measure unpaired electrons, providing an excellent approach to detecting ROS [28]. E171-aq samples were prepared in a 20 mg/mL stock and bath-sonicated for 16 min at 37 kHz. After bath sonication, 0.22 mg/mL E171 dilutions were prepared. Samples were immediately treated with 100 mM 5,5-dimethyl-1-pyrroline-N-oxide (DMPO) and analyzed. DMPO is a suitable spin trap for detecting hydroxyl radicals and other ROS-related radicals. For analyzing the samples, 100 µL of the sample was transferred into a 100 µL capillary (Brand, Eberstadt, Germany). The glass capillary was placed in the resonator of the ESR (Bruker, Karlsruhe, Germany, EMX 1273 spectrometer equipped with an ER 4119HS high-sensitivity resonator and a 12 kW power supply), which was operating at X band frequencies. The ESR measurements settings were as follows: 9.68 GHz power, 50.41 mW; modulation frequency, 100 kHz; modulation amplitude, 1 G; sweep time 41.94 s; time constant, 40.96 ms; sweep width, 50 G; and number of scans, 30. We determined the peak height in three biological replicates measured in technical triplicates and quantified them using WinEPR software (Bruker, Rheinstetten, Germany). E171-yog or control yogurt without E171 (Ctrl-yog) was diluted 1:3 (*v/v*) to make the samples viscose enough to be taken up by the glass capillaries, resulting in a final concentration of 0.22 mg of E171 per 1 mL of diluted yogurt. The yogurt samples were measured at the same settings as described above. To simulate mild inflammatory conditions and examine the behavior of E171 in an environment rich in ROS, we also measured all samples by adding 1 mM of H_2_O_2_, following a 30 min incubation in a 37 °C water bath. In inflammatory conditions, different types of ROS are formed, including H_2_O_2_ via, e.g., NADPH oxidases (NOX enzymes) present in various cells, particularly professional phagocytes and endothelial cells. These play an important role in the generation of the inflammatory response. These NOX enzymes, such as NOX4, DUOX1, and DUOX2, are prominent sources of H_2_O_2_ [29].

### 2.5. TIM-1 Model

The TNO GI Model of the stomach and small intestine (TIM-1) is a multi-compartmental dynamic in vitro digestion model used to study food products under physiological conditions. TIM-1 is designed to simulate the dynamic conditions in the lumen of the GI tract to predict the bioaccessibility of digestive effects and was previously described in detail [23]. In short, the natural GI system in humans consists of various organs, each of which is responsible for different digestion steps. These organs include the mouth, stomach, small, and large intestine. Food gradually passes through these different compartments during the digestive process, exposing it to various digestive enzymes and pH values. TIM is designed to simulate those dynamic processes in a computer-controlled and reproducible environment. TIM-1 consists of four compartments that resemble the stomach, jejunum, duodenum, and ileum, which are connected through peristaltic valve pumps that control food transport between the different sections. Before the samples were introduced into the TIM-1 model, they were prepared with artificial saliva, which includes electrolytes and alpha-amylase. Electrolytes, pepsin, and lipases drive gastric secretion. Duodenal secretion contains electrolytes, bile, and pancreatin. The pH was always controlled in the different compartments (Appendix A).

The system is equipped with hollow fiber membranes that remove digestion products and water, simulating uptake by the gut epithelial wall. In summary, TIM-1 can accurately reproduce the physiological conditions of the lumen of the upper GI tract by mimicking the secretions’ rate and composition and removing digested products and water. It is, therefore, a reliable tool for predicting bioaccessibility and the effects of GI digestion in vitro. TIM-1 provides high reproducibility due to its computer-controlled operating system. It has been extensively tested and can be compared to in vivo and human data, against which it has been validated [23,30]. All TIM-1 experiments were conducted as two biological replicates and measured in three technical replicates. The composition of all buffers and reagents utilized in the TIM-1 experiments can be found in the Appendix A 1.

### 2.6. Preparation of TIM-1 Samples

E171-aq and E171-yog were prepared as described in Section 2.2. To obtain the “meal”, 7 mL of E171-aq containing 140 mg of E171 or sterile MilliQ water was added to 150 g of gastric enzymatic solution (GES) and 148 g of citrate buffer. Additionally, we added 5 g of start–residue mixture. E171-yog was prepared by adding 155 g of E171-yog or Ctrl-yog and 5 g of start–residue mixture to 150 g of GES. Due to a limitation of the possible input volume of the sample in the TIM-1 model, the 210 g of yogurt used to prepare E171-yog was reduced to 155 g while including the estimated daily E171 intake of 140 mg/day, resulting in a final concentration of 0.9 mg of E171/per g of yogurt. E171-yog for the TIM-1 model was prepared as described in Section 2.2, except for a change in yogurt volume. The pH of those samples was adjusted to 4.5 by adding 1 M HCl or 1 M NaOH before incubation at 37 °C in a water bath for 20 min. After incubation, the “meal” mixtures for E171-aq and E171-yog or their respective controls (Ctrl-aq as sterile MilliQ water and Ctrl-yog, containing yogurt diluted with sterile MilliQ water equal to the amount of added E171 dispersion) were fed to the TIM-1 model. We sampled the ileal efflux (IE; material exiting the system at the end of the ileum) every hour for six hours. All IE samples were immediately snap-frozen in liquid nitrogen and stored at −80 °C/−20 °C until analysis. Samples aimed to be characterized by ESR were immediately treated with 100 mM DMPO, snap-frozen in liquid nitrogen, and stored at −80 °C.

### 2.7. Physicochemical Characterization of TIM-1 Samples

All meal and IE fractions (IE1-IE6) that were digested in the TIM-1 model are further referred to as digested E171-aq (dE171-aq) and digested E171-yog (dE171-yog), as well as their respective digested controls (dCtrl-aq, dCtrl-yog). The samples were analyzed using the spICP-MS protocol described in Section 2.3. The dE171-aq and dE171-yog samples were additionally characterized by measuring the hydrodynamic size distribution and zeta potential with a DLS from Malvern Nano ZS (Malvern Instrument, UK). All samples were diluted to a final concentration of 50 µg/mL using the cleared supernatant from the negative control samples of their respective IE fraction (IE1–IE6). Cleared supernatant fractions were obtained by centrifuging the respective IE fraction for 15 min at 15,000 rpm. Measurements for HD and the zeta potential were conducted at a concentration of 50 µg/mL of E171, with the temperature set to 25 °C, equilibration time of 0 s, viscosity of 0.8872 cP, and a refraction index of 1.330. E171-yog was analyzed using the same experimental conditions.

### 2.8. Measurement of ROS Formation via ESR Spectroscopy of TIM-1 Samples

For analyzing the digested TIM-1 samples, the same settings as in Section 2.4 were applied. The biological duplicates (n = 2) were measured in technical duplicates and analyzed using WinEPR software to determine the peak height (Bruker, Rheinstetten, Germany).

### 2.9. Analysis of Protein Corona Formation

To analyze the adhesion of protein to the digested E171 particles, we performed lithium dodecyl sulfate–polyacrylamide gel electrophoresis (LDS-PAGE) based on the protocols published by Ersöz et al. (2022) and Brouwer et al., 2024 [19,31]. In short, samples were centrifuged at 15,000 rpm for 15 min, and the supernatants were removed. The particle pellets were washed with 1× PBS and resuspended via pipetting, vortexing, and centrifugation at 15,000 rpm for 15 min. This step was repeated twice. The particle pellets were resuspended in 100 µL of 1x lithium dodecyl sulfate (LDS) sample buffer (Invitrogen, Waltham, MA, USA) and vortexed for 30 s. The samples were heated in a water bath at 70 °C for 10 min to denature the proteins bound to the particles. The samples were centrifuged at 15,000 rpm for 5 min to reduce the amount of titanium particles loaded on the SDS gel. Invitrogen 4–12% Tris-Bis gels (Invitrogen, Waltham, MA, USA) were loaded with 25 µL of protein–LDS suspension per well before the gel was run at 200 V for 47 min. Afterward, the gels were washed twice in MilliQ water and fixed in a fixation solution (10% glacial acetic acid, 40% methanol, and 50% water) for 20 min, followed by an additional washing step in MilliQ water for 10 min. The gels were stained with 0.1% Coomassie blue solution for 2 h before being destained overnight in MilliQ water on an orbital shaker set at 50 rpm. Gels were imaged via the Odyssey gel imaging system (Li-Cor, Homburg vor der Hoehe, Germany) at 700 nm fluorescence for Coomassie blue.

### 2.10. Statistical Analysis

All samples were measured in biological and technical triplicates if not stated otherwise. Normality was assessed via the Shapiro–Wilk-Test. Two-way Anova was performed to assess statistical significance with Bonferroni correction. Significance levels were set to * <0.05, ** <0.01, and *** <0.001.

## 3. Results

### 3.1. Particle Characterization in E171-aq and E171-yog

Following the NanoGenotox dispersion protocol, we determined the median particle size of E171-aq to be 78 nm and 79 nm spICP-MS and TEM, respectively. The percentage of particles in the nano-sized range (<100 nm) was 75% using spICP-MS and 73% using TEM analysis (Table 1). Figure 1A shows an exemplary TEM image of the particles displaying a spherical, round shape—Figure 1B shows the size distribution histogram indicating particles ranging from 40 to 180 nm in size. DLS measurements indicate E171 aggregates around 353 nm (PDI = 0.19) and a zeta potential of −31.4 mV (pH 7.2) at a concentration of 50 µg/mL of E171 (Appendix A).

E171-yog was characterized following the same procedure. Particle characterization with spICP-MS showed E171-yog particle aggregates with a median size of 330 nm and about 20% of particles being <100 nm (Table 1). TEM images in Figure 2A showed a strong background related to the biological matrix. The presence of the yogurt matrix, despite its dilution with sterile MilliQ water and relatively low concentration of E171, made the image analysis difficult, and no quantitative particle characterization could be performed. Exemplary images of E171-yog and the potential formation of a corona (black arrow) around the particles are shown in Figure 2A. Other structures attached to the particles might represent fat globuli; however, these have not been specifically analyzed in this work. Particle size distribution ranged from 60 to 650 nm (Figure 2B). DLS measurements showed aggregates above the detection limit and were deemed unreliable (Appendix A).

### 3.2. ROS Formation in E171-aq and E171-yog

ESR analysis with the spin-trap DMPO identified superoxide/hydroxyl radicals via a four-peak spectrum, as shown in Figure 3A. E171-aq significantly increased the formation of the DMPO^.^-OH signal level compared with the Ctrl-aq (MilliQ water). This ability slightly decreased when examining 0.22 mg/mL compared to 20 mg/mL, but E171 still showed a statistically significant ROS increase compared to the negative control (Figure 3B). When E171 was added to yogurt, the four-peak pattern was still visible; however, it was at a lower magnitude (Figure 3B). Quantifying the DMPO^.^-OH signal showed no difference in ROS levels when comparing Ctrl-yog to E171-yog. While there was a trend to a small increase in ROS formation in the E171-yog, the overall levels were close to the baseline. Furthermore, no additive effect was observed when E171 particles were stimulated with 1 mM H_2_O_2_, indicating that the E171 particles produced ROS without the involvement of transition metals.

### 3.3. Particle Characterization of TIM-1 Digested E171-aq and Digested E171-yog

We determined the effects of gastrointestinal digestion; hourly samples were obtained following dynamic in vitro digestion in the TIM-1 model for six hours. The further dilution of the initial input occurs within the TIM-1 model and its IE. The concentration and their dilution factor (DF) are displayed in Table 2.

Figure 4 shows the spICP-MS particle characterization of dE171-aq and dE171-yog following TIM-1 digestion for six hours. The digested negative controls, further referred to as dCtrl-aq and dCtrl-yog, were identified as blank samples despite low titanium background levels, resulting from the carry-over of E171 and calcium interferences from the yogurt (Appendix A). Digested E171-aq and E171-yog are further referred to as dE171-aq and dE171-yog, respectively. Our experiment showed that simulated GI digestion leads to the increased dE171-aq aggregation of 330 nm compared to its initial size characterization of 79 nm (E171-aq). E171-yog initially showed a size of approximately 330 nm, which decreased with GI digestion to aggregates of a slightly smaller size of 290 nm (dE171-yog). While dE171-aq increased in size following digestion in the TIM-1 mode, the opposite effect was observed for dE171-yog, which displayed slightly smaller aggregates after GI digestion, indicating a release of smaller aggregates from the yogurt food matrix.

While the percentage of nanoparticles fluctuated independently from the time point, they decreased in both conditions compared to the initial characterizations. The percentages of particles < 100 nm decreased from 72% to 1–7% for dE171-aq and from 20% to 1–5% for dE171-yog. These effects primarily occurred within the first hour of GI digestion and showed no significant changes over the following five consecutive hours of digestion.

### 3.4. ROS Formation Digested E171-aq and Digested E171-yog

ESR analysis showed that neither dE171-aq nor dE171-yog was able to induce ROS following GI digestion. The spectra in Figure 5 showed no DMPO-specific patterns for hydroxyl radicals, and the quantification of the signal did not show any ROS activity in the digested samples. Adding 1 mM H_2_O_2_, which mimicked inflammatory conditions in the intestine, increased ROS formation in all samples with increased time spent in the artificial digestion. This effect occurred for all control and E171 samples, indicating that the formation of ROS was related to the composition of the GI fluid and not the presence of E171 (Appendix A). The addition of E171 directly into IE fluids, without incubation within the TIM-1 model, did show the capacity of E171 to induce ROS, however, at much higher concentrations than when purely dispersed in MilliQ water. ROS formation only started at a concentration of 2 mg/mL (Appendix A).

## 4. Protein Corona

The evaluation of the protein corona around the dE171-aq and dE171-yog showed a time-dependent increase in protein attachment to the particles (Figure 6). All IE fractions showed proteins of 40 kDa and smaller bound to the E171 particles. The dE171-aq meal sample exhibited the lowest binding of proteins to the particles, while the sequential stages of in vitro digestion (IE1 to IE6) showed a gradual increase in protein attachment. LDS–gel electrophoresis analysis depicted distinct protein bands corresponding to various molecular weights, suggesting the presence of specific proteins with sizes ~40 kDa, ~37 kDa, 28 kDa, and ~18 kDa.

Similarly, dE171-yog exhibited a comparable pattern of protein binding to E171-aq. Additional protein bands were observed in the meal containing yogurt with bands at 70 kDa, 35 kDa, 34 kDa, 26 kDa, 24 kDa, and 18 kDa. IE1 showed additional bands at 35 and 34 kDa, like the dE171-yog meal.

Moreover, the formation of a protein corona was corroborated by TEM observations, providing supplementary indications of protein binding to E171 particles (Figure 2A). Our findings highlight the dynamic nature of protein corona formation during digestion and their accumulation over time. The accumulation of proteins attached to the particles over time also emphasizes that protein corona formation continues during gastrointestinal digestion and does not solely originate from the yogurt food matrix alone. Furthermore, the protein corona formation occurs dynamically over time, with protein bands increasing in size and occurrence.

## 5. Discussion

In this study, we aimed to evaluate the impact of a yogurt food matrix and in vitro GI digestion on the physicochemical characteristics and ROS formation properties of E171. Our study provides a comprehensive characterization of E171 particles under various conditions, including its pristine state in an aqueous environment representing a fasted state (E171-aq), within a yogurt food matrix (E171-yog), and during digestion in a validated artificial GI system. Additionally, we investigated the potential of E171 to induce ROS, specifically hydroxyl radicals, under these conditions and post-digestion. Moreover, we examined the protein corona formation on particles to gain insights into potential mechanisms influencing aggregation state and particle reactivity along the GI route.

The initial characterization of E171 via TEM revealed a considerable proportion of nanoparticles with a small size of 79 nm and 73–75% of particles < 100 nm and an HD of 335 nm. Incorporating E171 in a yogurt food matrix showed an increase in particle size to 330 nm and a reduction in nanoparticle percentage to 20%. Ferraris et al. (2023) have reported similar primary particle sizes using TEM, reporting a median Fmin diameter of 79 nm, 64% of particles < 100 nm, and an HD of 337 nm [16]. Verleysen et al. (2020) also tested a range of commercially available E171 on the market via spICP-MS. They showed a median particle size in a similar size range from 89 to 125 nm, with a nanoparticle fraction of 32–64%, highlighting the overall physicochemical properties of commercially available E171 [22].

Several studies, such as Mortensen et al. (2021), Marucco et al. (2020), and Dudefoi et al. (2021), assessed the size of E171 aggregates following digestion via DLS and reported aggregates ranging from 500 to 1000 nm on average [17,33,34]. Such large aggregates are of less physiological relevance given their size and reduced potential to be internalized by the intestinal cell layer since particles > 500 nm are mostly taken up via phagocytosis and macropinocytosis [35]. Our results showed an HD between 478 and 758 for dE171-aq and over 6000 nm for dE171-yog, with the size of dE171-yog particles exceeding the detection limit of the DLS machine. This highlights the issue of DLS measurements in complex matrices and simulated GI digestions, further supporting that DLS measurements are inadequate for accurately assessing the state of aggregation in such complex environments [16]. Our findings, along with those of Ferraris et al. (2023), demonstrated that when E171 samples were digested in an artificial GI system and analyzed using spICP-MS, the resulting aggregates that formed in the intestinal fluid were smaller. Ferraris et al. (2023) reported the sizes of digested E171 in aqueous dispersions between 178 nm and 259 nm, with a nanoparticle fraction ranging from 14% to 21%, and digested E171 in a fed state from 164 to 226 nm, with a nanoparticle fraction of 11–24%. Our study found that dE171-aq aggregates had a median size ranging from 300 to 355 nm, with a 1–7% nanoparticle fraction. Similarly, dE171-yog aggregates exhibited a median size ranging from 269 to 337 nm, with 1–5% particles smaller than 100 nm [16]. These findings suggest that E171 aggregates, particularly those from the dE171-yog, could still be internalized by intestinal cells via receptor-mediated endocytosis, as this mechanism accommodates particles up to 200–300 nm [35,36,37].

However, our study involved an extended digestion period, with the first sampling only occurring after one hour, which was twice the digestion time used in the Ferraris et al. (2022) study [16]. Despite this extended digestion period, we only observed slight changes in aggregation size or nanoparticle fraction over six hours. This suggests that the most substantial changes in aggregation and nanoparticle fraction occur within the first hour of in vitro gastrointestinal (GI) digestion. Therefore, it is crucial to closely monitor the behavior of particles during this initial hour to capture their dynamics and potential impacts on intestinal cells accurately. dE171-aq and dE171-yog fluctuated in their aggregate size during the various time points, which likely relates to the differences in E171 concentrations within the IE fractions and their ability to cluster together. Like Ferraris et al.’s (2023) samples in the fed GI state, dE171-yog showed overall smaller median aggregate sizes than the fasted state, indicating that a food matrix results in smaller aggregates than aqueous dispersions [16]. Both dE171-aq and dE171-yog demonstrated a reduction in nanoparticle percentage during digestion, showing the impact of GI processes on nanoparticle fraction, which is deemed particularly relevant regarding TiO_2_-associated adverse effects [38].

Our study assesses the ROS-inducing potential of E171 in food matrices. While E171-aq exhibited substantial ROS generation, this capability was suppressed upon mixing with a food matrix. This highlights a critical disparity in risk assessment practices, often focused on pristine materials in simple solvents and dispersions, which may not reflect real-world scenarios accurately. Previous work by Proquin et al. (2018) demonstrated the ability of E171 to induce ROS in cell-free environments, which was corroborated by our experiments [11]. ROS formation was entirely inhibited following GI digestion in the tested concentration. Marrucco et al. (2020) showed similar results using the DMPO spin probe [34]. These findings are crucial since the ability to generate ROS is determined to be an essential factor of E171 to induce adverse effects in vivo [1]. The inhibition of ROS formation following GI digestion might reduce the risk of E171-related adverse health effects. However, it is crucial to acknowledge that our findings only assessed the capabilities of E171 to induce ROS in a cell-free environment and did not capture potential interactions or effects of the digested E171 in a cellular system. These crucial interactions must be additionally considered when assessing the cytotoxic or genotoxic ROS formation effects of digested E171. Studies utilizing GI digestion with TiO_2_ nanoparticles and E171 showed associations with reduced cell viability, increased DNA damage, and chromosomal instability. These findings indicate that additional cyto- and genotoxic mechanisms were involved when digested E171 interacts with cells [19,20,21]. Findings on the DNA damage and genetic instability of TiO_2_ nanomaterials following GI digestion were of particular concern since those have been detected to cause effects at a physiologically relevant concentration of 14 µg/mL and an HD of 190 nm [20,21]. It is crucial to acknowledge that the size of said TiO_2_ nanoparticles remained small following this specific GI digestion and ranged below 200 nm. Particles ranging from 200 to 300 nm can still be internalized by cells, allowing direct interaction in the intracellular space and potentially with the nucleus and its DNA [6,16,35].

The formation of a protein corona during digestion contributes to the observed changes in particle behavior, as proteins and molecules from the digestive matrix adhere to the particle’s surface. This well-documented phenomenon significantly alters particle properties [19,34,39,40]. Our experiments showed a gradual increase in proteins attached to the particles over time. This increase in protein corona might contribute to the increased aggregation of particles as their electrochemical forces will pull the particles together and lead to a reduction in ROS generated by the E171 particles due to the coverage of their surface [16,34].

Our study underscores the importance of a food matrix and GI digestion and their modulation of particle properties, particularly size and ROS formation. Incorporating GI digestion into safety testing protocols is crucial for accurately assessing the potential risks associated with E171 exposure and ensuring human health protection. The present results help to bridge the gap between in vitro and in-human tests on the physicochemical properties of E171 and shed light on potential adverse effects in humans. The acquired insights in particle characteristics from this study are particularly relevant in the context of the recently finalized human dietary intervention on E171 at Maastricht University, which used the same E171-yog as an intervention product for the healthy volunteers participating in the study. The combined outcome of this in vitro study and the upcoming human study will help better understand potential adverse effects in humans following E171 ingestion. Furthermore, additional experiments of E171 digestion in the TIM-2 model for the large intestine are currently in preparation and will be published, highlighting the potential effects of E171 on microbiome composition, short-chain fatty acid production, and particle characteristics.

## 6. Conclusions

In conclusion, our study emphasizes the significant impact of a food matrix and GI digestion on the properties and reactivity of E171 particles. We observed significant alterations in particle size distribution, aggregation behavior, and ROS-inducing potential when E171 was introduced into yogurt as a food matrix and subjected to GI digestion. ROS formation was inhibited post-digestion, indicating profound alterations in particle reactivity, and this is potentially related to the formation of a protein corona. This could indicate that any potentially harmful effect of pristine E171 might be mitigated or reduced in vivo due to physicochemical alterations of the particles. These findings underscore the importance of incorporating dynamic GI digestion systems to mimic human-relevant physiological digestive conditions into the safety assessments of food additives or other ingested substances to accurately evaluate their potential health risks. These considerations are particularly important in the case of E171, contradicting scientific findings between in vitro and in vivo models, which eventually are caused by the lack of GI digestion in vitro. The discrepancy between findings from in vitro and in vivo studies is one of the main reasons that the safety of E171 is so controversially discussed. This is mainly because most in vitro do not use digested E171 in their experiments, therefore inaccurately mimicking the human situation while trying to provide hazard identification or hazard characterization for the human situation. The addition of prior in vitro experiments on digestion models might help minimize the discrepancies between the lines of experiments since they are closer to the human situation.

## Figures and Tables

**Figure 1 nanomaterials-15-00008-f001:**
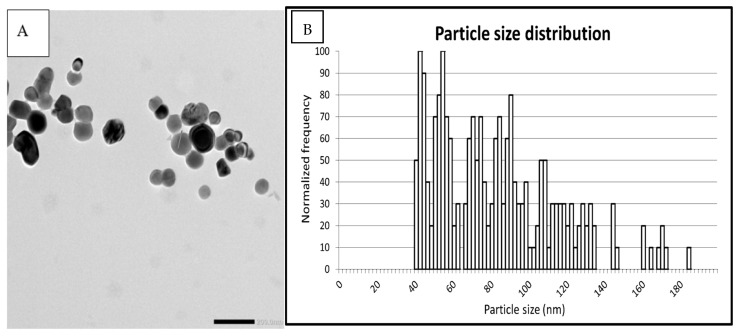
Particle characterization for pristine E171 following the NanoGenoTox dispersion protocol. (**A**) shows an exemplary TEM image of the particles, their form, and size, scale bar 200 nm, while (**B**) shows the normalized frequency of the size distribution of E171, ranging from 40 to 180 nm. Samples were prepared according to the standardized NanogenoTox dispersion protocol [32].

**Figure 2 nanomaterials-15-00008-f002:**
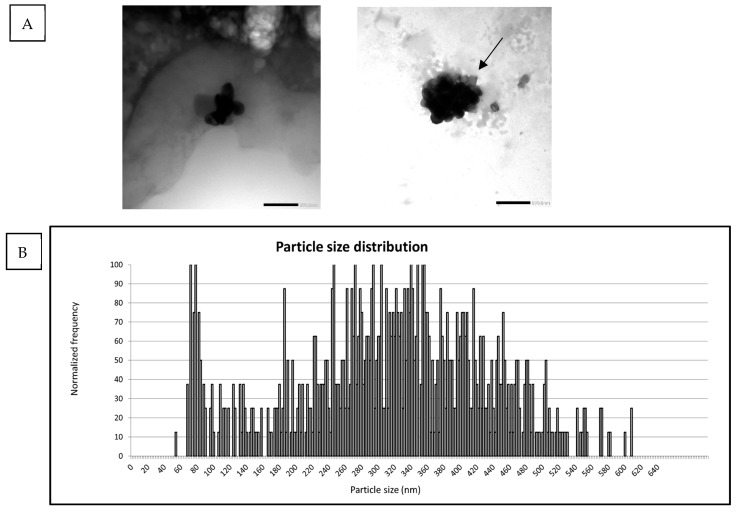
Particle characterization of E171-yog. (**A**) shows exemplary TEM images of E171 dispersed in yo-gurt. Due to the yogurt, it was not possible to obtain sufficient background to foreground image quality for the quantification of particle size and nanoparticle fraction. Nevertheless, these pictures show the increased aggregation state of E171, as well as the formation of a corona around the particles (black arrow). (**B**) shows the particle size distribution of E171 following spICP-MS meas-urement, ranging from 60 to 650 nm, further confirming an increase in particle/aggregate size in comparison to the pristine E171.

**Figure 3 nanomaterials-15-00008-f003:**
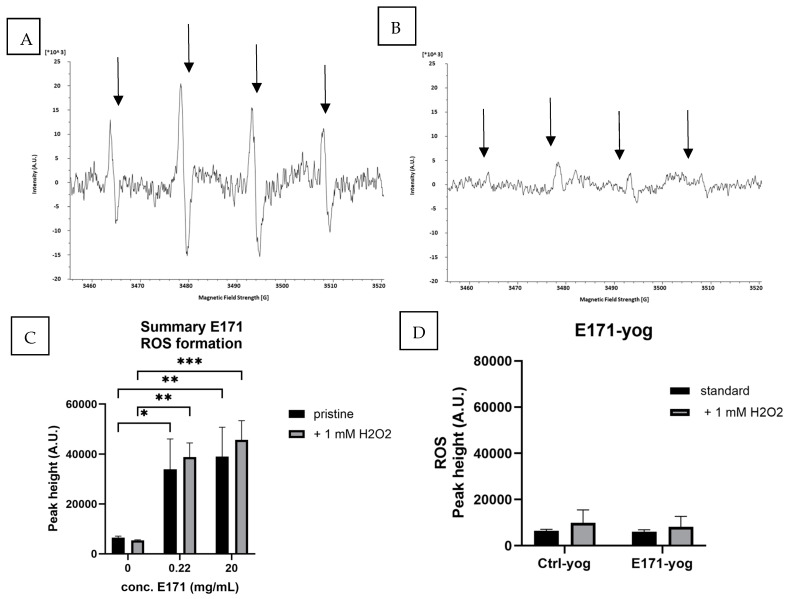
ESR spectra of E171-aq (**A**) and E171-yog (**B**) after incubation with the spin trap DMPO (100 mM). E171-aq was able to increase ROS formation significantly in comparison to Crtl-aq at 0.22 and 20 mg/mL (**C**). The presence of a food matrix reduces the ROS signal, showing an overall reduction in oxidative stress (**D**). Graphs present the quantified ROS levels in arbitrary units (A.U.). The addition of 1 mM H_2_O_2_ to mimic inflammatory conditions did not increase ROS levels. Significance levels are indicated by *p*-value * <0.05, ** <0.01, *** <0.001.

**Figure 4 nanomaterials-15-00008-f004:**
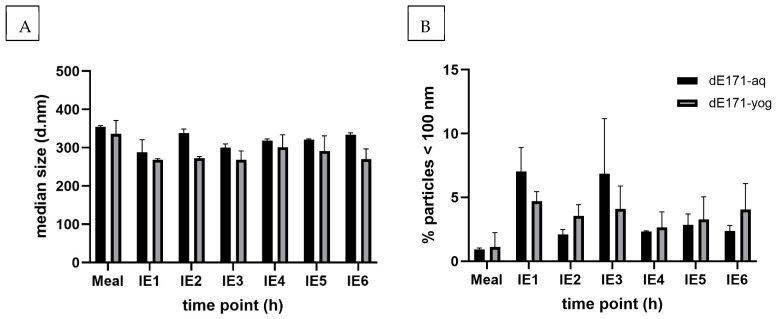
(**A**) Median size of digested dE171-aq and dE171-yog over six hours in the TIM-1 system. The particles tended to aggregate and formed stable aggregates at a mean of 330 nm for digested dE171-aq and 290 nm for dE171-yog samples. (**B**) Percentage of particles < 100 nm in the respective IE fractions. Nanoparticles ranged from 1 to 7% for dE171-aq and 1 to 5% for dE171-yog. Samples represent biological duplicates. IE = ileal efflux following 1–6 h of digestion within the TIM-1 model.

**Figure 5 nanomaterials-15-00008-f005:**
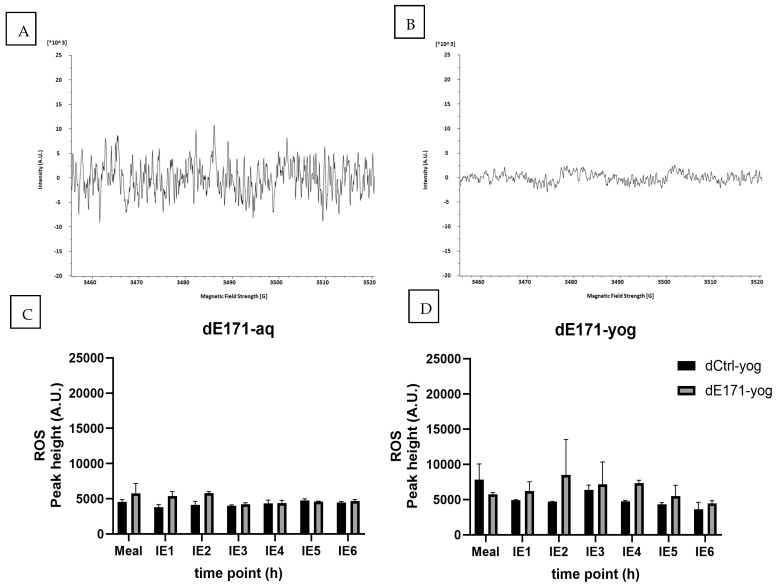
Evaluation of ROS via ESR spectroscopy for dE171-aq and dE171-yog, with their respective controls. (**A**) shows the spectra for dE171-aq, which no longer present the DMPO-specific 4-peak spectra, indicating no ROS presence in the sample. Quantification of the allocated peak position shows no ROS formation in comparison with dCrtl-aq. (**B**) shows the ESR spectra of dE171-yog, which does not show any ROS levels. Quantification of the spectra in (**C**,**D**) shows ROS levels near the baseline and no difference between E17 and Ctrl. Samples represent biological duplicates, measured as technical triplicates.

**Figure 6 nanomaterials-15-00008-f006:**
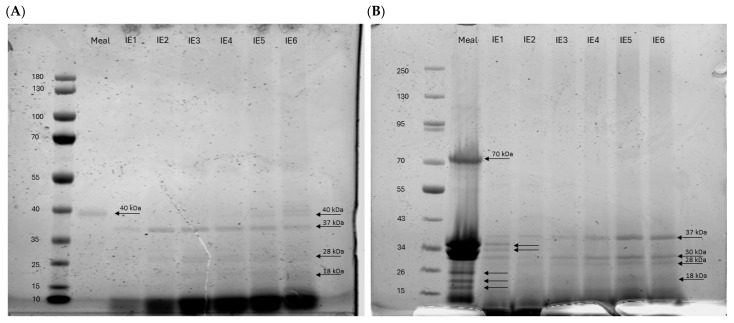
LDS-PAGE stained with 0.1% Coomassie blue. (**A**) shows the proteins attached to dE171-aq within the TIM-1 model over a period of six hours. The amount of protein attached to the particles gradually increased with time, as shown by the increased band intensity indicating higher protein binding. (**B**) shows the LDS-PAGE of dE171-yog, which exhibited a similar result compared to dE171-aq. Protein concentration also increased with time, and both samples showed specific bands at 37 kDa, 28 kDa, and 18 kDa.

**Table 1 nanomaterials-15-00008-t001:** Particle Characterization of E171-aq and E171-yog with spICP-MS and TEM.

Sample	Technique	Median Size (nm)	% of Particles < 100 nm
Pristine E171	spICP-MS	78	75
Pristine E171	TEM	79	73
E171-yog	spICP-MS	282	15
E171-yog	TEM	N/A	N/A

**Table 2 nanomaterials-15-00008-t002:** Calculated E171 concentrations and dilution factors (DF) for the respective IE (ileal efflux) fractions.

	Sample	Meal	IE1	IE2	IE3	IE4	IE5	IE6
Dilution factor	N/A	N/A	2.5	1.9	1.0	1.7	2.8	4.7
Volume (mL)	N/A	310	102	124	63.6	87.8	98.1	100.7
E171 (mg)	N/A	140	56.0	29.5	29.5	17.3	6.2	1.3
dE171-aq (mg/mL)	20	0.45	0.55	0.24	0.46	0.20	0.06	0.01
dE171-yog (mg/mL)	0.9	0.45	0.55	0.24	0.46	0.20	0.06	0.01

## Data Availability

The original contributions presented in this study are included in the article/Appendix A. Further inquiries can be directed to the corresponding author.

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
