# Peer review of "Investigating the ROS Formation and Particle Behavior of Food-Grade Titanium Dioxide (E171) in the TIM-1 Dynamic Gastrointestinal Digestion Model"

_nanomaterials, 2024, doi:10.3390/nano15010008_

Round 1

Reviewer 1 Report

Comments and Suggestions for Authors

The article is interesting, well written, however some minor comments are made.

How can the pH in the GI tract affect nanoparticles?

How does the food matrix of yogurt affect nanoparticles, apart from proteins, fats do not influence?

What then is the bioaccessibility of the samples?

Discuss what would be the mechanism of action of both the formation of ROS and the inhibition of ROS by the samples.

Author Response

Reply Reviewer 1:

Dear Reviewer, we thank you for taking the time to assess our manuscript and to provide us with your valuable remarks to help us improve the manuscript. Please find our responses listed below:

1) How can the pH in the GI tract affect nanoparticles?

Generally, the pH in the GI tract can affect particle aggregation and agglomeration, as seen after prolonged digestion in the GI tract within the TIM-1 model. The initial nanoparticle fraction of 78% particles <100 nm in the pristine material decreased to less than 10 % following digestion, with slight variability between the time points as seen in Figure 4. While this cannot solely be attributed to the pH alone since other components of the GI digestion system such as the feed were present as well, it still indicates the effect of pH in the particle aggregation state.Titanium nanoparticles (TiOâ‚‚ NPs) exhibit significant pH-dependent behavior regarding their size, aggregation, and colloidal stability. Their surface charge is governed by the protonation and deprotonation of hydroxyl groups, with the isoelectric point (IEP) typically falling between pH 5 and 7. Near the IEP, the zeta potential approaches zero, resulting in minimal electrostatic repulsion and increased aggregation tendencies. In contrast, at pH values below 4 or above 8, TiOâ‚‚ NPs exhibit greater stability and smaller hydrodynamic sizes due to stronger repulsion forces. This pH-dependent stability impacts their behavior in various environments; for instance, TiOâ‚‚ NPs are more dispersed under acidic gastric conditions but tend to aggregate in neutral or slightly alkaline intestinal environments. These findings highlight the critical role of pH in determining the stability and behavior of TiOâ‚‚ NPs in different systems.

2) How does the food matrix of yogurt affect nanoparticles, apart from proteins, fats do not influence?

Besides proteins, fats also accumulate around the particles and might contribute to the corona formed around the E171 particles. Figure 2 shows the corona formation around the particles, which consist of proteins and potentially also fats, while the more globular structure can be identified as potential fats attached to the particles. The presence of fat globuli structures were confirmed by our TEM expert Anna Undas who conducted an extensive analysis of hundreds of TEM images. We did not specifically characterize other attached molecules besides proteins to the surface of the titanium dioxide, but we agree that fats may be part of the corona of the particles. This is now added to Line 290-292.

3) What then is the bioaccessibility of the samples?

Usually, particles between 200-300 nm are still able to be taken up into the cell via endocytosis. As indicated in the text, our samples remain in this range and a potential uptake in colon cells is possible. This is mentioned in line 465 of the manuscript. Particularly, the particles digested in the yogurt matrix showed a lower particle size consistently ranging below 300 nm, which makes them candidates for endocytosis.

4) Discuss what would be the mechanism of action of both the formation of ROS and the inhibition of ROS by the samples.

ROS formation of E171 as detected by ESR indicated the formation of hydroxyl radicals, meaning that E171 scavenges an electron in an aqueous environment resulting in an unpaired electron at the oxygen molecule.  Different mechanism that can initiate this formation of ROS at the surface of the TiO2 particles is described in https://pubs.acs.org/doi/10.1021/acs.chemrestox.4c00235/ By adding the sample, the microenvironment seems to prohibit this interaction and the scavenging is not taking place anymore, at least not at the concentrations tested in this system. When analyzing ROS formation in only Ileal Efflux 6 with a high concentration of E171 as seen in the supplementary material Figure 3, it is still possible for E171 to produce ROS in a digestion environment, but at much higher concentrations of E171. We could speculate that this is happening in a dose-dependent manner, and eventually could also occur in a food matrix if the E171 concentration in food is high enough. However, we did not test this experimentally. For example, at 20 mg/mL, E171 produced the same amount of ROS in water compared to Ileal Efflux 6. However, it must be noted that the E171 in this experiment was directly added to the Ileale Efflux without undergoing prior digestion. The effect of the digestion on the formation of ROS is therefore not taken into account, nevertheless, the environment does allow for the production of ROS, when the E171 concentration is high enough. Lie 163-170.

Reviewer 2 Report

Comments and Suggestions for Authors

Bischoff et al. present a study on the effects of a simulated digestion on the pysico-chemical properties and ROS generating potential of E171 dispersed in either aqueous solution or yoghurt as relevant food matrix. With the TNO GI Model (TIM), the group used a sophisticated and well-established instrument to mimic a physiologically relevant passage through the gastric and small intestinal compartments. The group underlined the relevance of their study in the context of a related human dietary intervention study in which healthy volunteers were administered E171 in a yoghurt matrix as well.

Within this study the authors generated complementary results to previously published data on the effect of artificial digestive processes on E171. Especially in the context of the dietary intervention study a more detailed presentation of these results is warranted, even though it is a limitation of the study that “only” the processes of the stomach and small intestine were simulated. It remains unanswered how the passage through the large intestine – where the material would get in direct contact with the microbiome to interact with – would have affected the particles.

I have a couple of questions and comments before I would recommend this submission for publication.

The manuscript contains several misspellings and language mishaps, e.g. p. 2 “The interactopm between”, p. 2 “have implihigcations for particle”, p. 13 “Our studt assesses”, several in the conclusion.

Abstract “Food-grade titanium dioxide (E171) is widely used in food, feed, and pharmaceuticals for its opacifying and coloring properties.” This contrasts with the later information on its ban in food products in the European Union.

p.2 “Other studies showed a decrease in cell viability and increases in ROS formation and genotoxicity following exposure to physiologically rele-vant doses of various digested TiO2 nanomaterials in intestinal cell lines (20, 21). While these studies identified significant alterations to the size and aggregation status of E171, […].” The studies cited did not investigate E171. In general, the distinction between studies on E171 and studies on other TiO2 particles should be made more clearly and consistently.

p. 4 is the term “biological replicate” accurate in the case of cell-free / matrix-free samples?

Figure legends of Figures 1 and 2: please adapt to include information on the dispersant, i.e. aqueous solution and yoghurt, respectively.

Figure legend Figure 4: I know the abbreviation “IE” is introduced in the materials and methods section but it’s easily missed. Please add abbreviation again in the figure legend to facilitate readability.

In Figure legend 3 and section 3.4 the authors mention the use H2O2 for ESR measurements was to mimic inflammatory conditions. Please add one or two sentence to explain this approach.

p. 14 Conclusion: “These findings underscore the importance of incorporating dynamic GI digestion systems to mimic human relevant” While I agree that the consideration of digestive processes and a food matrix is relevant and important for the hazard assessment of materials incl. particles, I am not sure what the authors mean with “dynamic GI digestion”. Does this mean relevant information could only be drawn from a complex system like the TIM? For such a conclusion the study should have analysed the material in a more static digestive model as well.

“These considera-tions, are particularly important in the case for E171, contradicting scientific findingds between in vitro and in vivo models, which eventually are caused by the lack of GI diges-tion in vitro.” Can the authors please elaborate on such a strong statement? This should then also be discussed in more detail.

Supplementary information, Figure 1: Why is there such a strong development in the pH of the gastric compartment over the first 100 minutes?

Author Response

Reply Reviewer 2:

Dear Reviewer, we thank you for taking the time to assess our manuscript and to provide us with your valuable remarks to help us improve the manuscript. Please find our response listed below:

  • Only using the TIM-1 mode and not the TIM-2 model in combination.

We have a second manuscript in preparation assessing the effects of the TIM-2 model on E171, which assesses its interaction with the human microbiome and the production of short-chain fatty acids. However, the focus of this paper was to assess the state of aggregation of E171 when it arrives in the large intestine, where it supposedly invokes adverse effects in humans. For us, it was important to determine what size, and with what potential corona formation around the particles, the material arrives to establish a starting point for future risk assessment. Line 521-524

  • Misspellings

We apologize for the misspellings that were overlooked and double-check the manuscript again to avoid any further mistakes. All indicated errors were corrected and are highlighted in the revised text.

  • Abstract “Food-grade titanium dioxide (E171) is widely used in food, feed, and pharmaceuticals for its opacifying and coloring properties.” This contrasts with the later information on its ban on food products in the European Union.

This is a correct statement, while E171 is forbidden in the EU, the rest of the world, Asia, the Americas, and Africa, still use it at the currently allowed levels. However, the ban on E171 is only mentioned in the introduction. We added a sentence to highlight that the ban is only applicable in Europe and not the rest of the world.

Line 44-45: However, other countries such as the USA, the UK or China still allow it at dose levels currently considered safe.

  • Other studies showed a decrease in cell viability and increases in ROS formation and genotoxicity following exposure to physiologically relevant doses of various digested TiO2 nanomaterials in intestinal cell lines (20, 21). While these studies identified significant alterations to the size and aggregation status of E171, […].” The studies cited did not investigate E171. In general, the distinction between studies on E171 and studies on other TiO2 particles should be made more clearly and consistently.

We recognized the confusion and adapted the term to TiO2 nanoparticles, to make it clear to the reader that this study was solely performed on TiO2 nanoparticles and not food-grade TiO2. Line 72

  • 4 Is the term “biological replicate” accurate in the case of cell-free / matrix-free samples?

The experimental setup allows only one run of this experiment per day. Therefore the term biological replicate is applicable since these experiments were freshly set up every day with newly prepared materials and stock solutions for each of the conducted experiments.

  • Figure legends of Figures 1 and 2: please adapt to include information on the dispersant, i.e. aqueous solution and yogurt, respectively.

The samples were prepared according to the standardized Nanogenotox dispersion protocol. That was added to the description of the legends of Fig 1 and 2.

  • Figure legend Figure 4: I know the abbreviation “IE” is introduced in the materials and methods section, but it’s easily missed. Please add the abbreviation again in the figure legend to facilitate readability.

A definition of the IE1-6 was added to the figure description.

  • In Figure Legend 3 and Section 3.4 the authors mention the use of H2O2 for ESR measurements was to mimic inflammatory conditions. Please add one or two sentences to explain this approach.

In inflammatory conditions, different types of ROS are formed including H2O2 by e.g  NADPH oxidases (NOX enzymes) present in various cells, particularly the professional phagocytes and endothelial cells. which play an important role in the generation of the inflammatory response. These NOX enzymes, such as NOX4, DUOX1, and DUOX2, are prominent sources of H2O2 (https://pubmed.ncbi.nlm.nih.gov/38818374/). Line 163-170.s

  • 14 Conclusion: “These findings underscore the importance of incorporating dynamic GI digestion systems to mimic human relevant” While I agree that the consideration of digestive processes and a food matrix is relevant and important for the hazard assessment of materials incl. particles, I am not sure what the authors mean with “dynamic GI digestion”. Does this mean relevant information could only be drawn from a complex system like the TIM? For such a conclusion the study should have analysed the material in a more static digestive model as well.

We believe that the dynamic component adds a layer, which closer mimics the human situation since human digestion is a constant movement of the food throughout the GI tract. While this might be an optimal scenario, static digestion models like the INFOGEST digestion approach provide valuable and more easily accessible models that also allow to investigation of the effects of digestion, and therefore provide additional information. The combination of results from both experimental set-ups will help to better understand the effect of gastrointestinal digestion on E171 or other substances that undergo gastrointestinal digestion in general,

  • “These considerations, are particularly important in the case for E171, contradicting scientific findings between in vitro and in vivo models, which eventually are caused by the lack of GI digestion in vitro.” Can the authors please elaborate on such a strong statement? This should then also be discussed in more detail.

The discrepancy of finding between in vitro and in vivo studies is one of the main reasons that the safety of E171 is so controversially discussed. Mainly because most in vitro do not use digested E171 in their experiments, therefore not accurately mimicking the human situation, while trying to provide hazard identification or hazard characterization for the human situation. he addition of digestion models’ prior in vitro experiments might help to minimize the discrepancies between the lines of experiments since they are closer to the human situation. Line 534-540.